# Evolution and Application of Genome Editing Techniques for Achieving Food and Nutritional Security

**DOI:** 10.3390/ijms22115585

**Published:** 2021-05-25

**Authors:** Sajid Fiaz, Sunny Ahmar, Sajjad Saeed, Aamir Riaz, Freddy Mora-Poblete, Ki-Hung Jung

**Affiliations:** 1Department of Plant Breeding and Genetics, The University of Haripur, Haripur 22620, Pakistan; 2Institute of Biological Sciences, University of Talca, 2 Norte 685, Talca 3460000, Chile; Sunnyahmar13@gmail.com (S.A.); morapoblete@gmail.com (F.M.-P.); 3Department of Forestry and Wildlife Management, University of Haripur, Haripur 22620, Pakistan; sajjad.saeed@uoh.edu.pk; 4State Key Laboratory of Rice Biology, China National Rice Research Institute, Hangzhou 310006, China; aamirriaz33@gmail.com; 5Graduate School of Biotechnology & Crop Biotech Institute, Kyung Hee University, Yongin 17104, Korea

**Keywords:** genome editing, mutation, hybrid seed production, quality improvement, regulatory concerns, genetic gain, speed breeding

## Abstract

A world with zero hunger is possible only through a sustainable increase in food production and distribution and the elimination of poverty. Scientific, logistical, and humanitarian approaches must be employed simultaneously to ensure food security, starting with farmers and breeders and extending to policy makers and governments. The current agricultural production system is facing the challenge of sustainably increasing grain quality and yield and enhancing resistance to biotic and abiotic stress under the intensifying pressure of climate change. Under present circumstances, conventional breeding techniques are not sufficient. Innovation in plant breeding is critical in managing agricultural challenges and achieving sustainable crop production. Novel plant breeding techniques, involving a series of developments from genome editing techniques to speed breeding and the integration of omics technology, offer relevant, versatile, cost-effective, and less time-consuming ways of achieving precision in plant breeding. Opportunities to edit agriculturally significant genes now exist as a result of new genome editing techniques. These range from random (physical and chemical mutagens) to non-random meganucleases (MegaN), zinc finger nucleases (ZFNs), transcription activator-like effector nucleases (TALENs), clustered regularly interspaced short palindromic repeats (CRISPR)/associated protein system 9 (CRISPR/Cas9), the CRISPR system from *Prevotella* and *Francisella*1 (Cpf1), base editing (BE), and prime editing (PE). Genome editing techniques that promote crop improvement through hybrid seed production, induced apomixis, and resistance to biotic and abiotic stress are prioritized when selecting for genetic gain in a restricted timeframe. The novel CRISPR-associated protein system 9 variants, namely BE and PE, can generate transgene-free plants with more frequency and are therefore being used for knocking out of genes of interest. We provide a comprehensive review of the evolution of genome editing technologies, especially the application of the third-generation genome editing technologies to achieve various plant breeding objectives within the regulatory regimes adopted by various countries. Future development and the optimization of forward and reverse genetics to achieve food security are evaluated.

## 1. Introduction

It is estimated that approximately 800 million people around the globe are facing acute food shortages, and around 2 billion are facing nutrient deficiency [1]. Food and nutritional insecurity results in physical and mental impairment, reduced resistance to infectious diseases, and premature infant deaths [2]. This is complicated by the fact that the global human population is predicted to exceed 8.3 billion people by 2030 [3]. There is a need to significantly boost agricultural production by approximately 50% from current levels to ensure the availability of food [4]. To overcome these challenges and achieve the second of the United Nation’s Sustainable Development Goals, namely that of “zero hunger and improved nutrition,” intensive efforts are required to shift from traditional agricultural production systems to modern agricultural ones [5]. Plants are a basic source of food and energy, sustaining life on earth. A movement known as the green revolution began in the mid-20th century, where the use of agrochemicals and adoption of best agronomic practices contributed to optimized crop production along with traditional breeding techniques to develop semi-dwarf crop varieties with superior yield advantages [6]. However, the continuous application of agrochemicals has had serious negative environmental consequences. Future technologies must focus on mitigating these impacts and developing agricultural systems that are more resilient to climate stress. Improvements in sustainable crop production are essential to facilitate socio-economic development. Researchers have employed natural and induced mutations, heterosis breeding, and genetic manipulation techniques to promote sustainable crop production and enhance nutritional and food security [7]. Plant breeding and other technologies have made significant contributions toward minimizing hunger and extreme poverty over the last decades [8]. However, researchers have concluded that traditional breeding efforts alone cannot meet the ever-increasing demand for food for the human population [9]. Therefore, agricultural experts agree there is a need to convert to modern plant breeding approaches, especially novel plant breeding techniques (NPBTs) that are more flexible, reliable, and sustainable, to increase production in a way that does not impact negatively on natural resources [10].

The novel developments in agricultural biotechnology include engineering metabolic pathways that control traits of interest [11,12]. These have helped to develop crop plants with better agronomic benefits, nutrition, and resistance to both biotic and abiotic threats. These technologies complement the shortcomings of traditional breeding methods and are flexible, allowing for the use of genomic data of several models and non-model plant species. NPBTs depend largely on the public genome sequence database to target attributes of interest [13]. Traditional plant breeding tools and classical genome editing techniques (GETs) are unable to meet the demands of high precision, efficiency, and time constraints, guiding researchers to adopt NPBTs. The NPBTs include CRISPR/Cas9, CRISPR/Cpf1, BE, and PE. These techniques have proved to be powerful tools for the successful modification of genome sequences in a simple and precise manner [14]. GETs have been applied to several crop plants, and desirable phenotypes have been successfully obtained. NPBTs have enabled the production of transgene-free plants that are categorized as non-genetically modified (GM) crops. The NPBTs available can be exploited for various crop improvement programs to ensure nutritional and food security for increasing human population levels [15].

We have summarized the major developments in GETs in recent years, with a specific focus on the use of third-generation GETs for crop improvement programs. The discussion emphasizes yield and grain quality, hybrid seed production, and epigenetic modifications for the regulation of important traits in crops essential for food and nutritional security. Moreover, this review will enhance the understanding of speed breeding, omics, and precision breeding to achieve zero hunger, especially in developing countries. The discussion on the regulatory concerns of several countries regarding genome editing for crops and their derived products will also help to broaden the perspective of the scientific community; moreover, the integration of omics, speed breeding, and genome editing will foster our understanding to achieve genetic gain essential to meet global food demands.

## 2. Evolution of GETs

Traditional and modern crop improvement programs employing naturally existing or induced genetic variations require labor-intensive, time-consuming, and costly characterization of progenies for a series of generations derived from genetic crosses [16]. Researchers are developing new techniques to overcome the constraints of previous agricultural methods to meet the increasing demand for food [17]. NPBTs cover a broad range of techniques and are broadly categorized into three generations, although developments are still taking place to make GETs more user-friendly and efficient. Developments in NPBTs have contributed significantly to crop improvement programs compared to classical breeding efforts. NPBTs have the potential to modify endogenous genes to generate favorable phenotypes, similar to crop plants developed through traditional breeding [18]. The potential of NPBTs to enhance production has been documented in various crop species. A detailed comparison of the three generations of GETs is provided in Table 1, whereas Figure 1 depicts the timeline for the different developmental stages of GETs. This information can assist in decision-making about employing GETs in relation to specific objectives in crop breeding programs, especially the third generation. There are currently four families of engineered nucleases being used in genome editing, namely the engineered MegaN, ZFNs, TALENs, and the CRISPR/Cas9 nuclease systems [19,20]. With their potential applications for food security, we only discuss third-generation GETs in this review.

### 2.1. Third-Generation GETs

#### 2.1.1. CRISPR/Cas9 System

The CRISPR/Cas9, and related versions, are the most advanced third-generation GETs in plant biology [29]. The Cas9 system requires a short guide sequence (sgRNA) to direct Cas9 nuclease to cleave the target site [30]. Cas9 has the ability to cleave the double-stranded DNA target site complementary to sgRNA and successfully deploy various living backgrounds such as bacteria [31], eukaryotic cells (Cong et al., 2013), animal cells, mammalian systems [32,33], and plants [34].

#### 2.1.2. CRISPR/Cpf1 System

The development of a toolkit for genome editing through the addition of the class 2 CRISPR effector, Cpf1, has strengthened agricultural research [35]. The system from *Prevotella* and *Francisella*1 is known as Cpf1 but was previously known as Cas12a. This Cpf1 showed accuracy and efficiency in the genetic modification system, gaining researcher confidence [36]. The endonuclease of Cpf1 is comparatively smaller than Cas9 and therefore requires shorter CRISPR RNA (crRNA) with higher working efficiency [37]. The single RNA helps to bind Cpf1 upstream of the protospacer adjacent motif (PAM) and cuts DNA at a distance from the seed region, introducing five base pairs (bp) at the proximal end [35]. The Cpf1 system bypasses the trans-activating crRNA (tracrRNA) requirement for the maturation of crRNAs [38]. The Cpf1 system also efficiently manipulates the target site through T-rich PAM, whereas Cas9 technology requires a G-rich PAM sequence. It modifies the targeted region by keeping PAM sequencing intact based on its origin orthology [39]. The target-activated non-specific ssDNase activity, catalyzed by the same active site responsible for site-specific double-stranded DNA (dsDNA) cutting, is a fundamental property of CRISPR/Cpf1 enzymes. Moreover, the active nuclease site of Cpf1 cuts target single-strand DNA (ssDNA) in cis and the non-target ssDNA in transposition. This nuclease can only embed one DNA strand at a time, so the target and non-target DNA strands are presumably cleaved sequentially. This sequential cleavage of DNA elucidates the mechanism of staggered-end DNA break induced by Cpf1 [40]. There are several online tools, specifically the Cpf1-database (http://www.rgenome.net/cpf1-database/, accessed on 10 January 2021), which help to find the potential target site and design the gRNA in a fast yet easy way.

#### 2.1.3. BE System

A novel approach known as BE achieves more efficient genome manipulation with irreversibly based conversion at the target site. This technique is much simpler and more precise in nature, allowing the conversion of nucleotides without the formation of double-stranded breaks (DSBs) in the target DNA [41]. The change of cytosine (C) to thymine (T), called cytosine BE (CBE), demonstrated high efficiency [25,42]. The CBE system consists of four elements: (i) single sgRNA, (ii) dCas9, (iii) C deaminase, and (iv) uracil DNA glycosylase inhibitor (UGI). With an in-depth molecular understanding of deaminases, another system called adenine BE (ABE) was developed with the conversion efficiency of adenine (A) to guanine (G) [43,44]. The BEs restrict indel formation at both target and off-target sites without requiring DSBs for DNA modification [45]. This further allows single bp conversion; that is, bp substitutions without depending on donor DNA [41]. Recently, several BEs other than CBE and ABE have been developed, for example, rBEs (conversion from C to U). Moreover, another addition to GETs has taken place with the addition of a new technique called PE.

#### 2.1.4. PE System

The PE system allows manipulation of all 12 base-to-base conversions (transition and transversion), bypassing DSBs in targeted DNA [28]. The following technique utilizes Cas9 nickase bind with reverse transcriptase and PE guide RNA (peg RNA), consisting of a primer binding site (PBS), a target sequence, and a sequence to identify the target site. Hybridization of target DNA-pegRNA PBS and target DNA-reverse transcripts resulted in minimum off-target effects. To date, three generations of PE have been developed and categorized on the basis of their editing efficiency. First-generation PE (PE1) utilizes the mouse-murine leukemia virus (M-MLV RT), an RNA-dependent DNA polymerase, linked to the C-terminus of Cas9 nickase (H840A), which is an endonuclease with one inactivated domain. The efficiency of PE1 reached values of 0.7% to 5.5% when point transversions were introduced. The efficiency depended on the PBS length, and for different genes, various lengths of PBS (from eight to 16 nucleotides) resulted in higher efficiencies [46]. The efficiency of a second-generation PE (PE2) was enhanced, exhibiting a 1.6- to 5.1-fold improvement in the efficiency of introducing point mutations when compared to PE1. Furthermore, editing efficiency can be increased [28]. All 12 possible transition and transversion mutations were generated with 33% (±7.9%) efficiency in the PE3 system. The PE system has hampered the modification of promoter/introns more easily, allowing an allelic replacement at the target site to be feasible. It is noteworthy that the mutation efficiency of PE is similar to that of the BE system; however, the specificity was much higher than that of previously discussed GETs. The PE system is at the foundation stage, and further developments and applications for crop improvement programs will take place over time.

## 3. Application of GETs in Agriculture to Ensure Food Security

### 3.1. GETs for Crop Improvements

The challenge of food and nutritional security poses serious threats to human life and health, especially in developing countries. Over recent years, biotic (such as bacteria, insects, fungi, and viruses), abiotic (such as limited water supply, edaphic factors, heavy metal toxicity), and climatic (such as low and high temperatures, flooding, rainfall shifts) stresses have impacted negatively on crop production [47]. Based on the prevailing circumstances, researchers agree that traditional plant breeding methods alone cannot achieve a sustainable caloric supply to the expanding human population. Consequently, there is a need to switch to alternative cost-effective technologies with more flexibility and reliability to boost agricultural productivity with little or no pressure on non-renewable natural resources [10]. Current breeding methods focus on the increase in yield and yield-related traits per unit area to increase agricultural production. Thus, breeders play a key role in promoting agronomic traits to achieve economic gains [9]. Innovations in GETs have assisted in developing germplasms with improved characteristics and more accuracy over recent years.

#### 3.1.1. The CRISPR/Cas9 System–Proof of Concept for Crop Improvement

Improved traits in agriculturally important crops resulting from GETs, especially for yield and related traits, resistance to biotic and abiotic factors, and enhanced environmental resilience, are assisting in developing food security. The knockout of negative regulating loci, e.g., *GS3*, *DEP1*, *GS5*, *GW2*, *Gnla*, and *TGW6*, which control grain yield in *O.*
*sativa* L., resulted in a significantly improved grain yield in mutant plants [48]. Multiplex knockout of genes *GW2*, *GW5*, and *TGW6* resulted in a significant increase in the thousand-grain weight of rice grains [49]. Genetic manipulation of *OsERF922* resulted in the reduction in rice blast disease through pathogen infection [50]. Similarly, genome editing of the negative regulator gene, *Bsrk-1*, significantly reduced blast resistance without compromising yield [51]. The use of agrochemicals for crop production may cause serious environmental and human health-related impacts. Therefore, researchers are investigating herbicide resistance in crop plants [52]. The targeted manipulation of the *ALS1* gene controlling herbicide tolerance in rice had positive results [51], and the outcome of the investigation showed that the homology-directed repair (HDR) system was successful. Similarly, targeted mutagenesis in the second coding region of *BEL* in the *Japonica* rice cultivar, Nipponbare, showed resistance to the herbicides bentazon and sulfonylurea [53]. The seedling stage of rice is more prone to low-temperature stress, and the targeted modification of the transcription factor TIFY1b and gene *OsAnn3* significantly improved resistance to cold stress in mutant rice. To reduce heavy metal accumulation, Tang et al. [54] knocked out the *OsNramp5* transporter gene for cadmium (Cd), and the resultant mutant rice displayed a low accumulation of Cd in roots, shoots, and seeds.

Wheat (*T. aestivum* L.) provides caloric requirements to much of the human population worldwide. For disease resistance in wheat, the mildew-resistance locus *O* (*TaMLO*) gene was knocked out using the CRISPR/Cas9 system [55]. The mutant plants displayed resistance to powdery mildew disease caused by *Blumeria graminis* f. sp. *Tritici* (btg) s. Moreover, Gil-Humanes et al. [56] employed geminiviral-dependent DNA replicons in the wheat dwarf virus (WDV) to express Cas9 cassettes, which demonstrated a 12-fold increase in the expression of endogenous ubiquitin genes. This methodology and the promising nature of the results create opportunities for engineering complex genomes. Genetic manipulation of the wheat dehydration-responsive element-binding protein 2 (*TaDREB2*) and the wheat ethylene-responsive factor 3 (*TaERF3*) genes through protoplasts resulted in a 70% success rate for an improved response of mutant plants to abiotic stresses [57]. An efficient method for biolistic delivery in the host genome has been introduced to overcome the issue of transgene integration and off-target effects. This method allows for the delivery of ribonucleoproteins (RNP) in the targeted genome that degrades rapidly, allowing reduction in off-target effects. Liang et al. [58] used the same procedure for *TaGW2* and *TaGASR7* in two wheat varieties and recorded reduced off-target mutations in mutant plants. Transgene-free editing will help to circumvent strict regulatory measures and mitigate lengthy breeding procedures, for example, backcrossing to obtain transgene-free plants.

*Z. mays* is one of the leading cereal crops, and phytic acid constitutes approximately 70% of maize seeds. Liang et al. [59] knocked out *ZmIPK1A, ZmIPK*, and *ZmMRP4* to control phytic acid synthesis. The *AUXIN REGULATED GENE INVOLVED IN ORGAN SIZE* (*ARGOS*) gene family is a negative regulator of the ethylene response and signal transduction. The overexpression of the *ARGOS* gene displayed drought stress tolerance in mutant plants and the identification of novel allelic variants that can be further used in future maize breeding programs. The novel allelic variants of the *ARGOS8* gene, that is, *ARGOS8-v1* and *ARGOS8-v2*, were manipulated using CRISPR/Cas9. The resulting mutants were evaluated in multi-location trials. Mutants displayed a promising response compared to the wild-type under stress conditions [60]. Similarly, the phytoene synthase (*PSY1*) gene was manipulated using the U6 snRNA promoter, and the *psy1* mutant displayed white kernels and albino seedlings with no off-target mutations. Based on these findings, it can be assumed that the CRISPR/Cas9 system has been successfully employed for targeted mutagenesis of cereals. It is predicted that new developments in GETs can help overcome the limitations faced during genetic manipulation.

#### 3.1.2. The CRISPR/Cpf1 System–a Proof of Concept for Crop Improvement

The CRISPR/Cpf1 system has been employed for targeted mutagenesis of *Arabidopsis*, *O. sativa*, *Nicotiana tabacum, Glycine max, Z. mays, Citrus X sinensis*, and *Gossypium hirsutum*, etc. [61,62,63,64,65,66]. The CRISPR/Cpf1- and CRISPR/Cas9-mediated editing has been used for genome editing of the epidermal patterning factor-like 9 (*EPFL9*). The LbCpf1 system displayed a higher number of mutant T_0_ plants than the Cas9 system. The LbCpf1 system caused a 63 bp deletion compared to the deletion of 37 bp with the Cas9 system [67]. The Cpf1 system displayed 28.2% and 47.2% mutation rates in both tobacco and rice, respectively [61]. For targeted gene knock-in, both LbCpf1 and FnCpf1 endonucleases were used via the HDR system in plants. The results showed an 8% higher insertion efficiency in the LbCpf1 system compared to FnCpf1 in rice [68]. The Cpf1 system has displayed promising results and provides an alternative tool to edit the genome of both model and non-model plant species with more precision. However, there is a need to improve the GET toolkit to achieve greater precision, flexibility, and ease of handling. The BE system is an advanced method for genetic manipulation.

#### 3.1.3. The BE System–a Proof of Concept for Crop Improvement

The BE system was used to investigate the genetic mechanism of plant architecture and to determine how to enhance the efficiency of nutrient use through targeted mutagenesis of *SLR1* and *NRT1.1B* in rice. The mutant plants demonstrated a significant elevation in rice mutant plant height and nutrient use efficiency [69]. Similarly, Ren et al. [70] modified the binary vector by introducing pUbi:rBE3 and pUbi:rBE9 to target the genes *OsAOS1, OsJAR1, OsJAR2*, and *OsCOI2*, respectively, in rice. The results demonstrated that the rice base editor 9 (rBE9) resulted in higher editing accuracy and efficiency with lower off-target mutations compared to rBE3. The rBE9 efficiency increased owing to the presence of UGI from *Bacillus subtilis* bacteriophage PBS1, which stopped uracil N-glycosylase activity at the BE site [25]. Zong et al. [71] developed targeted mutagenesis through BE in two genes, *TaLOX2* and *ALS*, to develop herbicide-resistant wheat, rice, and potato, using a base editor fusion protein composed of Cas9 nickase and human APOBEC3A (A3A-PBE). The pnCas9-PBE and A3A-PBE showed higher efficiency in the conversion of C to T at the target sites of the genes under investigation. Based on the BE proof of concept, Tian et al. [72] successfully modified *ZmCENH3* and *ALS* gene editing in maize and watermelon. Similarly, a major gene, *TaALS-P174*, was targeted with a mutation efficiency of 75%. Mutant plants showed significantly increased tolerance to the herbicides imidazolinone, sulfonylurea, and the aryloxyphenoxy propionate-type [73]. The novel *G. hirsutum* BE 3 (GhBE3) introduced point mutations in *GhCLA* and *GhPEBP* genes controlling chlorophyll content and demonstrated a mutation frequency of 26.67 to 57.78% [74]. Li et al. [75] demonstrated the successful application of the ABE system for mutagenesis of *ACC*, *ALS*, *CDC48*, *DEP1*, *NRT1.1B*, and *OsEV*, resulting in a mutation efficiency of 7.5% in protoplasts and 59.1% in regenerated mutant rice and wheat plants. Moreover, an endogenous gene was also modified through a gain-of-function mutation, resulting in tolerant rice. Further application of the ABE system was evaluated for *IPA1* (*OsSPL14*), *OsSPL17*, *OsSPL18*, and *SLR1* genes and was effective in editing these genes through conversion of A to T and G to C in rice plants [76]. The BE (ABE7.8 and ABE7.10) use for *MPK6*, *MPK13*, *SERK2*, *WRKY45*, and *Tms9–1* genes showed significant on-target efficiency in mutant rice plants [77]. Jin et al. [78] analyzed the mutation efficiency of ABEs compared to CBEs in the *OsACC*, *OsALS*, *OsDEP1*, *OsNRT1*, *OsCDC48*, and *OsWx* genes in rice. The results demonstrated that the CBE system could be used to reduce off-target mutations. Recent developments in the BE system in plant species have been well documented in several independent studies [79,80,81]. The BE system has contributed significantly to elite germplasm development; however, there are fast-moving developments in GETs, and researchers are moving toward more reliable and easy techniques.

#### 3.1.4. The PE System–a Proof of Concept for Crop Improvement

Recently, the PE system has achieved indels from approximately 44 to 80 bp and point mutations with more precision and efficiency. Protoplasts in nine lines of rice and seven lines of wheat showed a mutation efficiency of approximately 19.2% [82,83]. Hybridization of target DNA-pegRNA PBS and target DNA-reverse transcripts resulted in minimum off-target effects. The application of GETs for crop improvement programs helped to develop germplasms with better yield, enhanced resistance to biotic and abiotic stresses and increased climate resilience (Figure 2). Improved crops can help to ensure food security. Developing NPBTs for crop improvement programs is also of interest to stakeholders, especially to feed growing human populations.

### 3.2. GETs for Hybrid Seed Production

Sustainable food production is challenging because of divergent cultural values, geographical boundaries, environmental factors, and technological differences. However, these difficulties can be resolved, and modern agricultural practices can be adapted to increase productivity per unit area [84]. NPBTs have the potential for effective use in heterosis breeding in agriculturally important crops. The development of hybrid seeds is a reality that has contributed significantly to increasing crop production and, ultimately, to income from farms, especially in underdeveloped countries. Hybrid vigor, exhibited in both plants and animals, allows hybrids to perform better than parental lines [85]. Hybrid seed production is achieved through three-, two-, and one-line systems. Each system has certain advantages and disadvantages. The prime importance of hybrid seeds is their higher yield and increased quality, enhanced resistance to biotic and abiotic stressors, and increased environmental resilience [86]. Three-line hybrid seed production systems have been less widely adopted owing to their laborious and time-consuming nature; therefore, two-line and one-line hybrid seed production systems are considered viable under current circumstances. However, the principles of the three hybrid development systems are relevant.

#### 3.2.1. First Generation Hybrid Development System

Male sterility (MS) is considered a worthy attribute for the efficient production of premium quality seeds. Therefore, MS systems have been studied and applied to several crop species. MS attributes have been classified into cytoplasmic male sterility and genic male sterility, based on the fertility of the gene source [87]. In the first generation (three-line) hybrid production system, fertility maintenance, and restoration are integral components [88]. Prof. Longping Yuan led a joint venture to identify and develop a commercially applicable hybrid production system during the 1960s. The researchers first identified wild male sterile rice varieties carrying the wild abortive cytoplasmic MS (*CMS-WA*) gene [89]. The *CMS-WA* gene has been introgressed into several rice lines to produce hybrid rice and is widely adopted in China. Hybrid rice has increased grain yield compared to inbred varieties [90]. In the first generation system, three lines, namely the CMS, maintainer, and restorer, require considerable time and labor to achieve, and commercial CMS limits the selection of parental lines. This influences the genetic diversity and restoration of CMS lines owing to cytoplasmic-nuclear interactions [91]. These limitations restrict the application of CMS systems for hybrid seed production and pave the way for the development and application of NPBTs for hybrid seed production that is more precise and cost-effective.

#### 3.2.2. Second-Generation Hybrid Development System

The two-line breeding system, known as second-generation hybrid development, depends on photo/thermosensitive genic MS (P/TGMS) lines under controlled conditions or maintainer lines under non-restricted conditions. The second-generation hybrid development system is convenient, as the manipulation of P/TGMS genes via GETs leads to the generation of MS plants [8]. Several genes controlling P/TGMS have been characterized and cloned in model species, such as rice. The first rice that has the photoperiod genetic male sterile (PGMS) gene was identified in 1973 and named Nongken58S. Nongken58S displayed complete MS characteristics under an extended photoperiod and restored fertility under a short photoperiod. Later, the PGMS character in Nongken58S was controlled via *pms1*, *pms2*, and pms3 [92]. The factor *pms3* encodes long non-coding RNA (IncRNA), which is long-day specific male fertility-associated RNA. A thermosensitive genetic male sterility (TGMS) *Indica* rice line was generated through the transformation of the P/TGMS gene from Nongken58S [93]. The TGMS attribute in *Indica* rice was controlled by *p/tms12–1*, encoding a small RNA13 of 21-nucleotide nucleotides. Carbon-starved anthers (CSA) are associated with reverse photoperiod-sensitive genic MS (rPGMS) rice during short photoperiods and are fertile during long photoperiods. CSA encodes the MYB transcription factor R2R3, which mediates sugar partitioning. Moreover, UDP-glucose pyrophosphorylase1 (Ugp-1) splicing depends on temperature, and overexpression of Ugp-1 causes TGMS in rice [94]. The first *Indica* TGMS Annong S-1 (AnS-1) rice was identified in 1987. Zhou et al. (2014) studied the *TMS5* gene and found that the gene encodes endonuclease RNase ZS1 in AnS-1. The endonuclease RNase ZS1 degrades the temperature-sensitive ubiquitin fusion ribosomal protein L40 (UbL40), which influences TGMS characteristics. The *tms5*-dependent rice line plays an important role in two-line hybrid development [95]. Moreover, further studies identified the presence of a mutation in *TMS5* from 24 of 25 commercial MS lines. Barman et al. [8] and Zhou et al. [96] knocked out the *TMS5* gene and successfully developed a TGMS line for hybrid seed production. In wheat, identification of the *Ms1* gene has provided a platform for novel hybridization strategies. The generation of induced biallelic frameshift mutations in *Ms1* resulted in the complete MS wheat cultivar Fielder and Gladius. These selected non-transgenic MS lines helped to produce hybrid wheat. The successful application of Cas9 for P/TGMS-related genes has provided many options for employing Cpf1, BEs, and PEs. The newly available GETs can reduce various negative characteristics, for example, off-target effects and low mutation rates, and also have the ability to generate transgene-free plants in greater numbers. The advent of multiplex genome editing has paved the way to use more sophisticated techniques to achieve the desired objective within minimum time. The simultaneous knockout of multiple genes in a single vector construct has helped to generate a multi-control sterility system.

#### 3.2.3. Multi-Control Sterility (Third-Generation) Hybrid Development System

Third-generation hybrid development, known as the multi-control sterility (MCS) system, uses the transgenic method to develop hybrid seeds. The gene *ZmMs7* was isolated through fine-mapping and functional characterization and encoded a transcription factor PHD-finger orthologous to *PTC1* in rice and *MS1* in *Arabidopsis*. The MCS was used to develop a transgenic maintainer line that can be deployed for wheat hybrid seed production. The *ms7-6007* transgenic maintainer line was developed through the transformation of the MCS vector construct consisting of (i) the *ZmMs7* gene to restore fertility, (ii) α-amylase gene *ZmAA*, (iii) the DNA adenine methylase gene *Dam* to devitalize transgenic pollen, (iv) the red fluorescence protein gene *DsRed2* or *mCherry* to mark transgenic seeds, and (v) the herbicide-resistant gene *Bar* for transgenic seed selection. The transgenic maintainer line is self-pollinated and later produces red fluorescent (transgenic) and normal color seeds (non-transgenic) at a ratio of 1:1. Moreover, the *Japonica* male sterile mutant, *ms26/ms26*, was developed through the transformation of male fertile mutants, *Ms26, Zm-aa1, and*
*DsRed2*, through rice optimized codons. *Ms26 encodes* cytochrome P450 mono-oxygenase, which further catalyzes ω-hydroxylation of C16 and C18 fatty acids in the tapetum [97]. A large population of maintainer lines has been developed and characterized at both phenotypic and molecular levels. The best perming line (full fertility restoration, single-copy transgene, 1:1 segregation with viable and non-viable pollen, 1:1 segregation of transgenic and non-transgenic seeds) was selected to enhance environmental and food safety. A novel nuclear MS, *O. sativa No Pollen 1 (OsNP1)* gene was identified through positional cloning. *OsNP1 encodes* glucose-methanol-choline-oxidoreductase, which is essential for tapetum degeneration and pollen exine formation [98]. A novel MS rice mutant, *Osnp1*, was developed through the ethyl methanesulfonate mutagenized gene in the Huanghuazhan cultivar. Similarly, a binary vector was developed containing two separate T-DNAs, namely the *NPTII* gene with CaMV 35S promoter and another second T-DNA containing the *OsNP1* gene with a native promoter, *Zm-aa1* with pollen-specific *PG47* promoter, and *DsRed with* aleurone layer-specific *LTP2* promoter. Both of these T-DNAs were transformed to *the osnp1* mutant, and screening and characterization of several transformations led to the identification of a single T_1_ Zhen18B. The mutant containing the second T-DNA was selected as the maintainer line and displayed normal vegetative and reproductive growth. The selfing of Zhen18B produced a segregated population with transgenic (fluorescent) and non-transgenic (non-fluorescent) plants in a 1:1 ratio. The mutant plants with fluorescence were similar to the Zhen18B maintainer, whereas the non-fluorescent plants were similar to *the osnp1* mutant. Thus, selfing and selection among the Zhen18A population provide maintainer lines for commercial hybrid seed production. Zhen18A was used as a female parent, and cross-pollination was performed with approximately 1200 paternal lines. Of the hybrids produced, 85% displayed a higher yield than the parents; however, 10% were transgressive segregants. The outcome of the breeding program showed the promising nature of third-generation hybrid seed production. Zhen18A was recently approved by the Crop Variety Appraisal Committee of Guangdong Province. All three generations of hybrid seed development systems are described in Figure 3.

#### 3.2.4. Induced Apomixis through Genome Editing to Preserve Heterosis

Heterosis breeding has contributed significantly to improving crop production and ultimately income from farming. However, the segregation of traits in subsequent generations forces farmers to buy costly seeds for each sowing season. NPBTs have addressed this problem through de novo modification of genes controlling sexual reproduction to apomixis, which has been successfully performed in *Arabidopsis* [99]. Khanday et al. [44] knocked out three genes, *BBM1*, *BBM2*, and *BBM*, which caused embryo arrest and abortion, and the wild-type attribute (fertility) recovered through the male-transmitted *BBM1*. These findings indicate that fertilization during embryogenesis is mediated by the pluripotency factors transmitted from the male genome. The conversion of mitosis for meiosis (*MiMe*) phenotype, through genome editing, is combined with the expression of the *BBM1* gene in egg cells to obtain clonal progeny, preserving genome-wide parental heterozygosity [100,101]. The induced (synthetic) apomixis is heritable in multiple generations of clones and is therefore known as a clonal *fix* strategy. Wang et al. [102] edited *REC8*, *PAIR1*, and *OSD1* meiotic genes by multiplexing from hybrid rice, producing clonal diploid gametes and tetraploid seeds. Similarly, Kelliher et al. [103] and Li et al. [104] manipulated the *MTL* locus responsible for fertilization through the CRISPR/Cas9 system and obtained haploid maize hybrid seeds. The editing of endogenous genes, *OsSPO11-1, OsREC8, OsOSD1*, and *OsMAT*, resulted in the *MiMe* phenotype [105]. The application of GETs for the preservation of hybrid vigor in rice proved to be a proof of concept for possible use in other crops, ultimately lowering the cost of production. However, the number of viable clones with intact heterosis is limited. To date, only 30% of seeds with intact F_1_ properties have been reported in the F_2_ generation. Underlying pathways controlling the *MiMe* phenotype to increase the percentage of seeds with intact hybrid vigor in F_2_ should be explored further. Moreover, the efficiency and accuracy of genomic alternation of genes controlling the MiMe phenotype can be increased through the use of BEs and PEs. The application of GETs for crop improvement and hybrid seed production has greatly enhanced the average yield per hectare. However, several countries have adopted regulatory regimes limiting their global application.

## 4. GETs for Improved Grain Quality

Improvement in the quality of grain is a key attribute for plant breeders. It is a quantitative trait that is simultaneously influenced by various factors, including environmental ones. In recent years, breeding efforts for semi-dwarf varieties and heterosis have contributed significantly to achieving high yield, but with low quality, and this aspect has been the subject of much research [106]. The availability of genome sequencing data for several model and non-model species has facilitated novel gene identification, targeted genome modification, and functional characterization of genes controlling grain quality traits. To date, little success has been achieved through the application of genetic markers to identify the genomic regions controlling grain quality-related traits. Researchers are now using NPBTs for large-scale and rapid evaluation of traits in various plant populations. The GETs have contributed significantly to the development of premium quality cultivars within a short time period.

### GETs Proof of Concept for Grain Quality Improvement

The hybrid rice (*Indica*), grown in mainland China, has a high amylose content (AC), which makes the grains dry and hard when cooked. The AC is controlled mainly by the *Waxy* (*Wx*) gene. Ma et al. [107] manipulated the *Wx* gene through the CRISPR/Cas9 system in the *Japonica* cultivar, and the mutant displayed reduced AC. Moreover, Zhang et al. [108] and Li et al. [109] used the same genome editing system to generate functional mutations in *Japonica* cultivars “Xiushui134” and “Wuyunjing 7”. These mutations helped to reduce AC mutants without compromising agronomic traits. The genetic factors *SBEI* and *SBEII* play an effective role in determining the physical properties and the fine structure of starch. The CRISPR/Cas9 system was employed to manipulate both *SBEI* and *SBEII*, which displayed the underlying role of *SBEII* in the creation of high AC rice. The *BADH2* gene is responsible for controlling the fragrance in rice through the deposition of substrate 2AP. The 8-bp mutation in the *BADH2* gene resulted in a higher accumulation of 2AP substrate in the resulting genotypes [110]. Rice contains six 5-methylcytosine (5mC) DNA methylase genes (*OsROS1, OsROS1b, OsROS1c, OsROS1d, OsDML3a*, and *OsDML3b*) that contribute significantly to the nutritional quality of the grains. It is possible to manipulate these genes through GETs to develop germplasms with improved nutritional quality (Table 2). The storage protein in wheat, gluten, can cause health issues in consumers, such as celiac and non-celiac disease and gluten ataxia. The gene controlling α-gliadin was knocked out through the CRISPR/Cas9 system. Screening for transgene-free and off-target mutagenesis found that the mutants with reduced gluten content could be further used for breeding low-gluten wheat cultivars [111]. Targeted mutagenesis of *GW2* controlling RING-type E3 ubiquitin ligase assisted in developing an understanding of the regulatory mechanism of cell numbers in spikelet hulls in increasing the crude protein content of wheat, together with the increased weight of grains [112]. RNA interference (RNAi) and the CRISPR/Cas9 system were employed to manipulate the *ZmMADS47* gene by controlling a MAD-box protein interacting with O_2_ to activate the zein gene promoter. The mutant developed through RNAi showed a zein content of 16.8%, whereas the MADS/CAS9-21 mutant showed a zein content of 12.5% [113]. The disruption of the *Wx1* gene controlling granule-bound starch synthase through the Cas9 system generated several versions of *Wx* mutants, which can be further used for various purposes, especially crossbred as CRISPR–waxy hybrids [114]. Researchers at DuPont successfully manipulated the *ARGOS8 gene* to develop drought-resilient maize. The knocking of the native *GOS2* promoter into the 5′ untranslated region *ARGOS8* generated maize mutants with better yield under water-limited conditions.

*A third-generation* base editor (BE3), APOBEC1-XTEN-nCas9-UGI, was employed in rice to test its feasibility and efficiency. Three targets were chosen: one target (P2) in *OsPDS*, which encodes a phytoene desaturase, and two targets (S3 and S5) in *OsSBEIIb*, which encode a starch branching enzyme IIb in rice. We delivered the vectors into rice calli through *Agrobacterium*-mediated transformation. The results displayed precise point mutations at three target sites in rice, thus providing a feasible and effective tool for targeted BE to improve nutritional quality [109]. Similarly, an efficient A·T to G·C BE system was employed in rice. The A·T to G·C mutation resulted in the desired amino acid substitution or potential interference of miRNA binding in the target regions of the *Wx* gene. The mutation frequency induced by the pHUN411-ABE vector was <10% at the *Wx* and *GL2* targets [141]. To date, the CRISPR/Cas9 system has been widely deployed for improvement in the quality of the grain, but novel Cas9 variants (e.g., Cpf1, BE, and PE) have immense potential for improving grain quality. These variants demonstrate superior results with less chance of off-target effects and higher numbers of transgene-free plants.

## 5. Multiplex Genome Editing for Complex Traits

Metabolic pathways are responsible for economically important traits in plants. These metabolic pathways are controlled by complex genetic networks within cellular systems. Therefore, molecular techniques with the ability to handle several loci are worthy of both basic and applied research [142]. GETs allow the genetic manipulation of several genes through multiplexing, that is, editing multiple target sites [143]. Multiple gRNAs were assembled in the Golden Gate cloning or Gibson Assembly method, driven by different promoters [144]. Xie et al. [145] employed a simple strategy to engineer endogenous tRNA through a simple, efficient method of editing multiple loci using the CRISPR/Cas9 system. The CRISPR/Cpf1 system has a dual nuclease that cleaves targeted DNA and its own CRRNA [146]. Wang et al. [65] demonstrated the feasibility of multiplex editing in rice using the Cpf1 system. The metabolic engineering of *OsGSTU*, *OsMRP15*, and *OsAnP* responsible for the transport and accumulation of anthocyanin were mutated simultaneously in a rice line with purple leaves to generate green leaf mutants [107]. Similarly, Ma et al. [107] targeted three sites on the *OsWaxy* gene in rice and generated a mutant with an AC content reduced from 14.6% to 2.6%. Li et al. [147] edited five genes in the tomato plant that controlled the carotenoid biosynthesis pathway, using six gRNAs matching two targets in the *SGR1* gene and one each in the genes *LCY-E*, *Blc*, *LCY-B1*, and *LCY-B2*. All of these accumulated more lycopene than wild-type tomato plants. *GW2*, *GW5*, and *TGW6* in rice were edited simultaneously with three different gRNAs to introduce simple indels via the NHEJ pathway. Genes controlling grain weight have also been targeted [49]. The multiplex editing system helped to knock out various genes, for example, *Brassinosteroid Insensitive 1*, *Jasmonate-Zim-Domain Protein 1*, and *Gibberellic acid insensitive* in *Arabidopsis* and Rice Outermost *Cell-specific gene 5*, *Stromal Processing Peptidase*, and *Young Seedling Albino* in rice, and successfully obtained the desired phenotypes [148]. Multiplex editing has been actively used to induce mutations in numerous loci in plant genomes and is considered a reliable tool for precise genome modification

## 6. Challenges and Future Perspectives

The revolution in the field of molecular biology and the discovery of the CRISPR sequence in the microbial immune system has allowed biotechnologists to induce mutations in any genome of interest with specificity and efficiency. These NPBTs have provided scientists with the ability to achieve the precise and speedy manipulation of desirable traits compared to conventional breeding methods. Advancements parallel to GETs provide valuable opportunities to exploit existing genetic resources to develop crop varieties with premium yield, high nutrition, and resistance to biotic and abiotic stresses. Although GETs have several advantages over classical plant breeding protocols, they also face challenges in their application in agricultural crops. Molecular-level studies are challenging in non-model plant species because of the difficulty in identifying loci controlling important traits [149]. Genome sequencing in non-model crops has enabled researchers to identify the genes controlling important phenotypes. Plant species lacking reference genome can be target sequenced using degenerate primers to predict the putative function of traits of interest (Figure 4).

### 6.1. Regulatory Concerns Regarding Genome Editing for Crops and Derived Products

The debate on GM and genome editing for crops require governmental intervention to formulate clear and uniform regulatory policies. Although the Cartagena Protocol on Biosafety advanced an understanding of the international trade of GM organisms and plants, many governments have divergent opinions on development, commercialization, production, and consumption thereof [150]. Presently, genome editing for crops falls within the ambit of two regulatory guidelines, i) process-based and ii) product-based [151,152]. Moreover, the regulation of genome editing for crops varies between countries. Some nations deal with genome editing for crops as GM, while others deal with such crops as non-GM [151]. For instance, the governments of the United States of America and Brazil have agreed to regulate genome editing for crops in a similar manner to those developed by conventional breeding [114]. The Canadian regulatory guidelines state that any plant-based technology aimed at developing new attributes must comply with the Canadian Food Inspection Agency regulations [153]. The Court of Justice of the European Union (ECJ) has declared that crops produced via NPBTs must be regulated in the same way as GMOs. However, traditional mutagenic techniques with established biosafety records are exempted [154]. To ensure adequate risk assessment and management, the State Council of China has formulated the “Regulation on Administration of Agricultural Genetically Modified Organisms Safety” and has categorized genome editing with GM crops [155]. Similarly, the Indian, Japanese, and New Zealand regulatory bodies categorize genome editing for crops as similar to GM and apply strict biosafety guidelines [156,157]. Regulations to deal with genome editing for crops are largely dependent on the existing regulatory framework within a particular country. The advancement in GETs to produce transgene-free plants may assist in circumventing enforced biosafety-related regulations followed for conventional transgenic plants [152]. In summary, it is the responsibility of all stakeholders to debate the regulatory framework further and to develop uniform regulations that promote the safety of humans, animals, plants, and the environment.

### 6.2. Transgene-Free Breeding

The advent of transgene-free breeding has highlighted numerous options for targeted genetic modification without genome disorders [158]. Organisms modified through DNA-free editing are considered non-GMOs in the traditional understanding of plant biology and biotechnology [159]. There has been a focus on three approaches for Cas9/gRNA delivery to achieve DNA-free editing. The most popular method is the delivery of an in vitro assembled RNA. A variation of this approach is the formation of more complex nanostructures that are non-identical to virus-like particles [160]. Nanoparticles allow the delivery of premade protein-RNA complexes and the incorporation of mRNA and gRNA for successful expression of Cas9, followed by the assembly of Cas9/gRNA in the plant cell and subsequent transgene-free editing [161]. Moreover, nanoparticles allow for improvements in cargo stability and delivery efficiency. The second approach is to employ a virus-mediated delivery of encoding RNA templates. Engineering viruses with the CRISPR/Cas system for transgene-free plant genome editing is a significant challenge because of restrictions related to viruses. However, the delivery of the whole CRISPR/Cas9 system in plants has only recently been made possible, using virus vectors. Ma et al. [162] successfully delivered the complete CRISPR/Cas9 cassette in *Nicotiana benthamiana*, thus obtaining transgene-free genome-edited plants with sufficiently high efficiency. The in vivo processing approach developed by Cody and Scholthof [163] might be the key to designing novel DNA-free editing methods. The third approach is the most intriguing. It is an implementation of the *Agrobacterium tumefaciens* type IV secretory system for Cas9 delivery as a protein into plant cells. Through the application of these delivery systems, the DNA-free editing approach has been successfully applied to a number of species. A review by Metje-Sprink et al. [164] reports that researchers have achieved transgene-free editing in *N. benthaminiana* [162], *Solanum tuberosum* [165], *T. aestivum*, and *Z. mays* [166,167], *Brassicaceae* [168], *O.*
*sativa* [169], *Musa acuminata* [170], *Lactuca sativa* [171], and *Piper nigrum* [172]. However, this method has some drawbacks. The first is the low editing efficiency compared to other delivery approaches. This may be improved by engineering the bacterial delivery mechanisms. Nevertheless, the use of *Agrobacterium* for DNA-free delivery of at least the protein component is a fascinating achievement with great potential. Its use to deliver base editors for gene editing is yet another confirmation of new possibilities in DNA-free editing [173]. It is noteworthy that in all reported species, transgene-free editing has resulted in inheritable modifications, regardless of the delivery system used. The ability to generate transgene-free plants can help circumvent strict regulatory regimes adopted by several countries for genome editing of crop plants.

### 6.3. Off-Target Effects

CRISPR/Cas9 is a state-of-the-art technology, and the targeting specificity of Cas9 is believed to be tightly controlled by the 20-nt guide sequence of the sgRNA and the presence of a PAM adjacent to the target sequence in the genome. However, potential off-target cleavage activity could still occur on DNA sequences with even three to five bp mismatches in the PAM-distal part of the sgRNA-guiding sequence. The high frequency of off-target activity (≥50%) of RNA-guided endonuclease-induced mutations at sites other than the intended on-target sites is a major concern [174]. Cas9 specificity is much higher in bacteria (small genome size) than in eukaryotes (large genome size). Cas9 in bacteria has evolved without selection pressure; there is thus a high chance of off-target effects in a genome larger than the bacteria [175]. So far, different strategies, such as GC content, gRNA length, truncated gRNA, and chemical modification, have been developed to reduce off-target effects. Along with these methods, computational models for the selection of optimal DNA targets and the corresponding sgRNAs have displayed minimum off-target effects. However, the development of computational efforts requires a more extensive database for different experimental conditions, including different cell types and species. Additionally, Cas variants, for example, BE and PE, are also critical for reducing off-target effects [28,43]. The ever-increasing developments in GETs can not only reduce the off-target effect but also increase the on-target efficiency.

### 6.4. Genetic Gain Through Speed Breeding

NPBT allows researchers to use gene bank accessions and mutant collections for gene discovery and deployment. Speed breeding reduces the number of cycles required to produce crop varieties. The extended photoperiod and controlled temperature regimes for rapid generation cycling in fully enclosed glasshouses for large-scale application in crop breeding programs are used. Under traditional varietal development procedures, a 2% genetic gain (2050 food demand challenge) is a huge challenge for numerous reasons, such as a narrow genetic base, low harvest index, and a lack of elite breeding stock, especially in developing countries with dense populations [176]. The genetic gain was calculated using the following equation:ΔG=i×h×σA/L

***i*** = selection intensity

***h*** = square root of narrow-sense heritability

σA = square root of additive genetic variance

***L*** = length of breeding cycle

***L*** holds immense importance in achieving genetic gains through the introduction of novel desirable alleles through rapid breeding cycles. Four to seven generations per year have been reported in various crop species, such as wheat, durum wheat (*Triticum turgidum*), barley (*Hordeum vulgare*), chickpea (*Cicer arietinum*), pea (*Pisum sativum*), and canola (*Brassica napus*) [177]. Moreover, the vegetative growth period was successfully achieved by establishing short days when growing maize and rice to trigger the reproductive stage under greenhouse conditions. The promising nature of speed breeding not only helps in the study of the genetic aspects but also the introgression of favorable alleles into elite germplasm. Moreover, genomic selection has been demonstrated as a promising breeding strategy to accelerate genetic gain for heterosis breeding [178,179,180]. The large-scale genotyping of breeding material through SNP chips and next-generation sequencing has enabled easier and more cost-effective genomic selection. The integration of these novel developments holds the ability to achieve the genetic gain objective within minimum time and with more precision.

## 7. Conclusions

In view of rapid developments in agriculture, especially with respect to plant breeding, there is a need to develop an integrated mechanism for the use of these technologies in service to humanity. The use of GETs with speed breeding can greatly reduce the duration of the breeding cycle, and omics generated data can enhance the efficiency of identifying genes and their potential role in pathways controlling traits of significance. The identified genes can be knocked in and/or out through GETs, ultimately promoting precision in plant breeding. Recent developments in GETs, such as BEs and PEs, have resolved several concerns raised by regulatory bodies. The generation of transgene-free plants is helpful in categorizing genome editing of plants developed through classical breeding methods. The generation of transgene-free plants has significantly increased in BEs and PEs, and the application of speed breeding for genome editing material can further increase the number of transgene-free plants within a short period of time. Recently, the application of nanoparticles has taken place for efficient vector delivery into the host genome. The nanoparticles provided safe, efficient, and direct cytosolic/nuclear Cas-RNPs delivery in any type of cell, with lower off-target mutation compared to plasmid-based CRISPR systems; however, food security concerns in developed and developing nations must be central to the research. However, there is still a quest to develop more efficient, reliable, and cost-effective systems to develop germplasm as per human requirements. In conclusion, global food security must be based on innovations taking place in the present to meet future needs. It also requires the development of a framework based on lessons learned. Therefore, to exploit the full potential of NPBTs, a multipronged approach is needed that encompasses technology development, dissemination of information, adoption of research outcomes, and social acceptance of the product. It is stated with confidence that NPBTs are powerful enough to resolve the global hunger crisis, and the global scientific community must exploit this opportunity by developing NPBT user-friendly regulatory frameworks and support mechanisms.

## Figures and Tables

**Figure 1 ijms-22-05585-f001:**
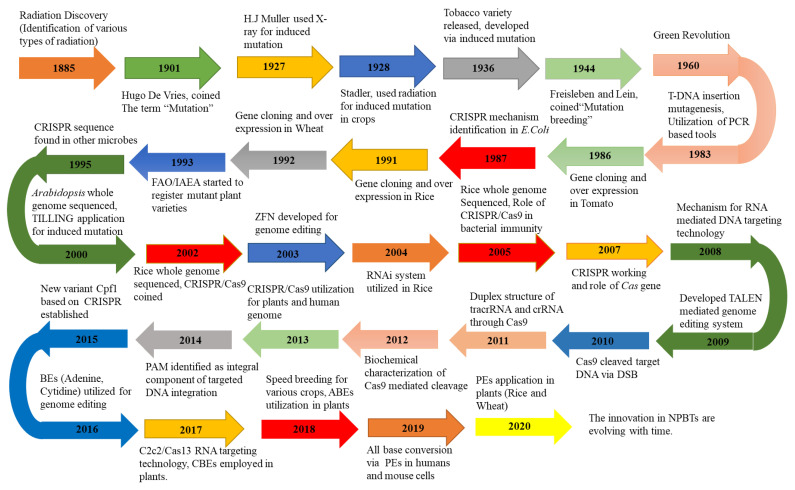
A brief history of different versions of GETs, showing historical events. CRISPR = clustered regularly interspaced short palindromic repeats; Cas9 = CRISPR-associated protein system 9; Cpf1 = CRISPR system from *Prevotella* and *Francisella*1; BEs = base editors; PEs = prime editors; T-DNA = transgenic deoxyribonucleic acid; PCR = polymerase chain reaction; FAO = food and agriculture organization; IAEA = international atomic energy agency; DSB = double-stranded breaks; RNAi = RNA interference; tracrRNA = trans-activating CRISPR RNA; PAM = protospacer adjacent motif; NPBTs = novel plant breeding techniques.

**Figure 2 ijms-22-05585-f002:**
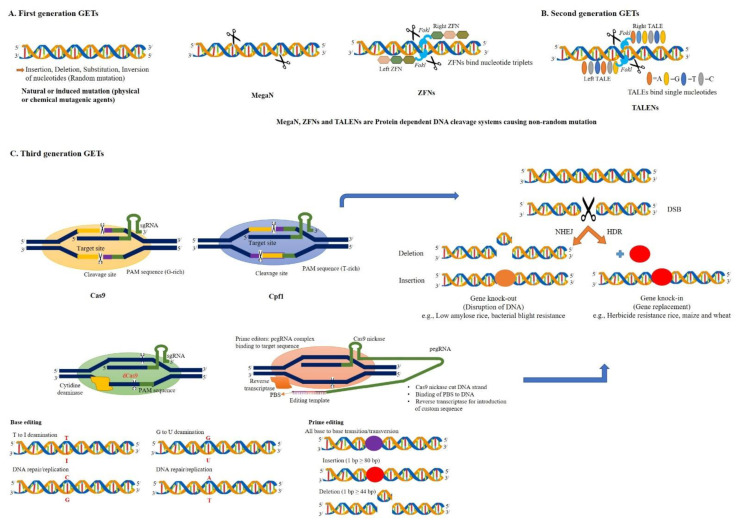
Schematic models of gene editing systems. (**A**) The first generation included induced mutation through mutagenic agents, Radiations and EMS (Ethyl methanesulfonate), meganuclease (MegaN), and Zinc finger nucleases (ZFNs); (**B**) second generation included transcription activator-like effector nucleases (TALENs); (**C**) third generation included the CRISPR-associated protein system 9 (Cas9), the CRISPR system from Prevotella and Francisella1 (Cpf1), BE and PEs. The GETs from MegaN, ZFNs, TALENs, CRISPR/Cas9, Cpf1 generate DSBs. BE and PE create mutations without DSBs. DSB = double-stranded breaks; dCas = catalytically inactive (dead) Cas; PAM = protospacer adjacent motif; PBS = primer binding site; NHEJ = non-homology end joining; HDR = homology-directed repair.

**Figure 3 ijms-22-05585-f003:**
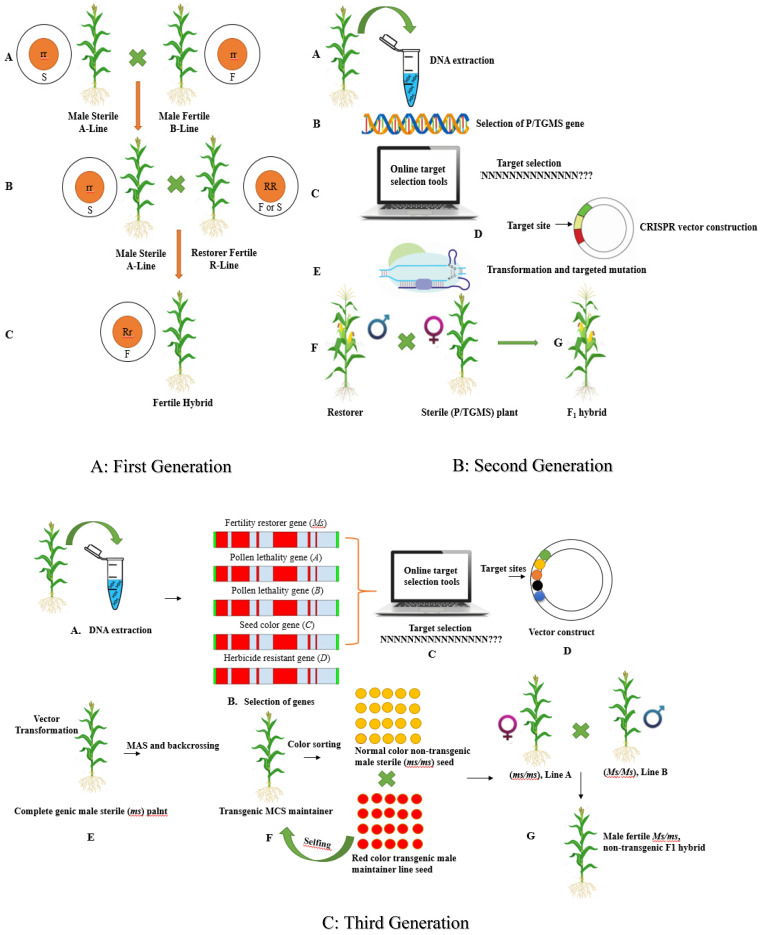
Schematic description all three generations of hybrid development. (**A**) First generation or three-line hybrid development system consisting of A (male sterile), B (male fertile), and R (restorer) lines; (**B**) second-generation or two-line hybrid development system through targeted mutagenesis of the P/TGMS gene, later crossed with restorer line; (**C**) third generation or MCS system through multiplexing of genes controlling MS, pollen lethality and color sorting. The mutants are backcrossed with selfing of the MCS maintainer line to develop desirable plants.

**Figure 4 ijms-22-05585-f004:**
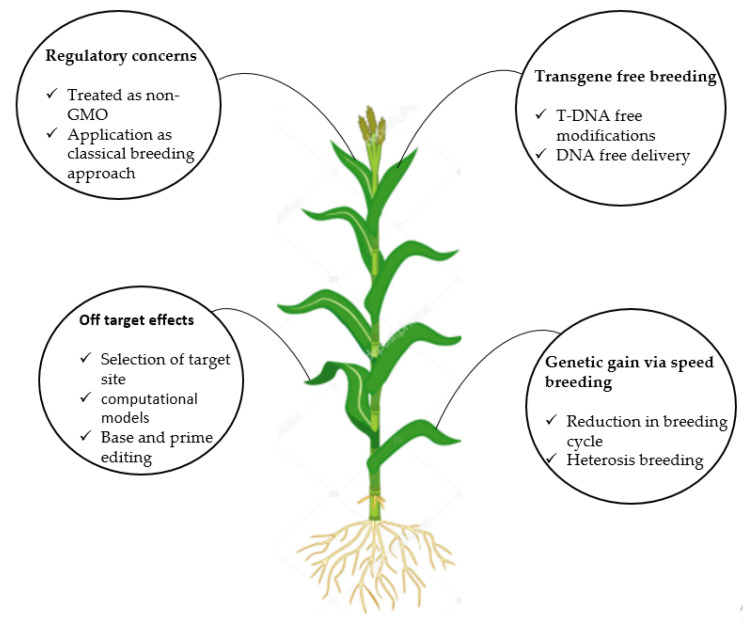
Future perspectives of novel plant breeding techniques for genetic modification in plants genome.

**Table 1 ijms-22-05585-t001:** Comparison of MegaN, ZFN, TALEN, CRISPR/Cas9, Cpf1, base editing, and prime editing.

Property	MegaN	ZFN	TALEN	CRISPR/Cas9	CRISPR/Cpf1	BE	PE
DNA binding determinant	Meganuclease	Zinc finger protein	Transcription-activator-like effector	CrRNA/sgRNA	CrRNA/Cpf1	dCas/nCas	nCas9/pegRNA
Recognition	Protein-DNA	Protein-DNA	Protein-DNA	RNA-DNA	RNA-DNA-Protein	RNA-DNA-Protein	RNA-DNA-Protein
Endonuclease	Meganuclease	*FokI*	*FokI*	*Cas9*	*Cpf1*	*dCas*	*pegRNA*
Mutation rate	High	Medium	Medium	Low	High	High	Very High
Target size length (bp)	14–40	18–36	30–40	22	20–24	4–6	8–15
Off-target effects	High	High	Low	Variable	Low	Low	Very low
Mechanism of action	Able to induce double-strand breaks (DSB) with two possibilities of Non-homology end joining (NHEJ) and homology-directed repair (HDR), depends on the designing tool						No DSBs
Design feasibility	Difficult, may require substantial efforts to design engineered protein	Required customized protein for each gene sequence. Oligomerized pool engineering (OPEN) used to select for new zinc finger assays	Technical challenging due to repeating sequence. Golden gate molecular cloning used to produce a TALE array	Easy to clone, only 20nt to targeting each gene expressed in a plasmid.	Easy	Easy	Easy
Multiplexing	Not possible	Difficult	Difficult	Easier	Easier	Easier	Not tested yet
Methylation sensitivity	High	High	High	Low			
Target recognition efficiency	Low	High	High	High	High	Very high	Very high
Cost-effectiveness	No	No	Moderate	High	High	Very high	Very high
Application	Human, Animals, and Plants	Human, Animals, and Plants	Human, Animals, and Plants	Human, Animals, and Plants	Human, Animals, and Plants	Human, Animals, and Plants	Human, Plants (rice and wheat)
References	[21]	[22,23]	[22,23]	[22,23]	[24]	[25,26,27]	[28]

CrRNA = CRISPR RNA; sgRNA = single-guide RNA; dCas = catalytically inactive (dead) Cas; nCas = nickase Cas; pegRNA = prime editing guide RNA; bp = base pair; MegaN = meganuclease; ZFN = zinc finger nuclease; TALEN = transcription activator-like effector nucleases; CRISPR = clustered regularly interspaced short palindromic repeats; Cas9 = CRISPR-associated protein system 9; Cpf1 = CRISPR system from *Prevotella* and *Francisella*1; Bes = base editing; Pes = prime editing.

**Table 2 ijms-22-05585-t002:** List of genes edited through the application of CRISPR/Cas9, Cpf1, base editing, and prime editing systems in plants.

Specie.	GET System	Trait of Interest	Gene Function	Target Gene	Transformation Method	Reference
*Oryza sativa* L.	CRISPR/Cas 9	Yield and quality improvement	Increases length and yield	*OsPPKL1*	Agrobacterium	[115]
			A key enzyme of aromatic amino acids biosynthesis	*EPSPS*	Biolistic transformation	[116]
			Regulators of inflorescence Architecture of plant height	*DEP1*	Agrobacterium	[48]
			High amylose	*SBEIIb*	Electroporation	[117]
			Amylose content	*Waxy*	Agrobacterium	[108]
			Isoamylase-type debranching enzyme	*ISA1*	Agrobacterium	[118]
			Negative regulator of thermosensitive genicmale sterility	*TMS5*	Agrobacterium	[96]
			Low phytic acid	*OsITPK6*	Agrobacterium	[119]
			Enhanced fragrance	*Badh2*	Agrobacterium	[110]
			Grain weight	*GW2,*	Agrobacterium	[49]
			Grain weight	*TGW6*	Agrobacterium	[49]
			Grain weight	*GW5,*	Agrobacterium	[49]
			Early maturity of rice varieties	*Hd2,*	Agrobacterium	[120]
			Early maturity of rice varieties	*Hd4*	Agrobacterium	[120]
			Early maturity of rice varieties	*Hd5*	Agrobacterium	[120]
			Improved growth and productivity	*PYLs*	Agrobacterium	[121]
		Biotic stresses	Various abiotic stress tolerance and disease resistance	*OsMPK5*	Agrobacterium	[122]
			Rice blast resistance negative regulator	*ERF922*	Electroporation	[50]
			Resistance to rice tungrospherical virus	*eIF4G*	Agrobacterium	[123]
			A key enzyme for the biosynthesis of branched-chain amino acids (major targets for herbicides)	*ALS*	Agrobacterium	[124]
			Salinity tolerance	*OsRR22*	Agrobacterium	[125]
			Various abiotic stress tolerance and disease resistance	*OsMPK5*	Agrobacterium	[122]
		Nutritional improvement	Low Cd-accumulation	*OsNramp5*	Agrobacterium	[54]
			Potassium deficiency tolerance	*OsPRX2*	Agrobacterium	[126]
			Low cesium accumulation	*OsHAK-1*	Agrobacterium	[127]
	CPf1	Yield and quality	Grain length-yield	*OsGS3*	Agrobacterium	[66]
			Leaf and yield	*OsDEP1*	Agrobacterium	[64]
			Grain yield	*OsNAL*	Agrobacterium	[66]
			Floral organ identity	*OsDL*	Agrobacterium	[61]
			Negatively modulates bulliform cells	*OsROC5*	Agrobacterium	[64,128]
		Abiotic stress	Carotenoid biosynthetic pathway	*OsPDS,*	Agrobacterium	[109]
			Herbicide resistance	*OsALS*	Agrobacterium	[78]
			Abscisic acid regulation-stress tolerance	*OsNCED1*	Agrobacterium	[61]
			Caroteniod catabolism and abscisic acid metabolism-stress tolerance	*OsAO1*	Agrobacterium	[61]
			Abiotic stress tolerance	*EPFL9*	Agrobacterium	[67]
			Herbicide resistance	*OsBEL*	Agrobacterium	[65]
			Herbicide resistance	*OsRLK*	Agrobacterium	[65]
	BEs	Yield and quality	Amylose content	*OsWaxy,*	Agrobacterium	[129]
			Spikelet and floral organ	*SNB*	Agrobacterium	[130]
			Grain shape	*SLR1,*	Agrobacterium	[130]
			Male fertility	*Tms9-1,*	Agrobacterium	[130]
			Grain weight	*OsSPL14,*	Agrobacterium	[130]
			Grain size	*OsSPL17,*	Agrobacterium	[130]
		Biotic stress	Rice blast resistance gene	*Pid3*	Agrobacterium	[131]
		Nitrogen transport and leaf death	Nitrogen transport	*OsACC1,*	Agrobacterium	[130]
			Nitrogen transport	*OsNRT1,*	Agrobacterium	[78]
			Leaf senescence	*OsCDC48,*	Agrobacterium	[78]
*Triticum aestivum*	CRISPR/CAS9	Yield and quality	Grain weight negative Regulator	*TaGW2*	Biolistic transformation	[58]
			Low-gluten	*Alpha-gliadin*	Biolistic transformation	[111]
			Control grain length and weight	*TaGASR7*	Biolistic bombardment	[132]
		Biotic stress	Mildew-resistance locus	*TaMLO*	Agrobacterium	[133]
			Powdery mildew-resistance negative regulator	*TaMLO-A1*	Biolistic bombardment	[134]
			Disease resistance against powdery mildew	*TaEDR1*	Biolistic transformation	[135]
		Abiotic stress	Fe content	*TaVIT2*	Biolistic bombardment	[136]
	BEs	Yield and quality	Control grain size and weight	*TaGW2*	Agrobacterium	[75]
			Inflorescence architecture and affects panicle growth and grain yield	*TaDEP1,*	Agrobacterium	[75]
		Biotic and Abiotic stress	repress resistance pathway to powdery mildew	*TaLOX2*	Particle bombardment	[137]
			Herbicides resistance	*TaALS,*	Particle bombardment	[138]
	PEs	Yield and quality	Control grain length and weight	*TaGW2*	Agrobacterium	[82]
			A gibberellin regulated gene that controls grain length	*TaGASR7*	Agrobacterium	[82]
		Biotic stress	Repress resistance pathway to powdery mildew	*TaLOX2*	Agrobacterium	[82]
			Mildew-resistance locus	*TaMLO,*	Agrobacterium	[82]
*Zea mays*	CRISPR/Cas9	Yield and quality	45 (male sterility)	*MS45*	Biolistic-mediated transformation	[139]
			Increased grain yield under drought stress	*ARGOS8*	Agrobacterium	[60]
			Phytoene synthase	*PSY1*	Agrobacterium	[140]
			Seed and leaves traits	*ZmIPK1A,*	Agrobacterium	[59]
			Seed and leaves traits	*ZmIPK*	Agrobacterium	[59]
			Seed and leaves traits	*ZmMRP4*	Agrobacterium	[59]
		Abiotic stress	A key enzyme for the biosynthesis of branched-chain amino acids (major targets for herbicides)	*ALS2*	Agrobacterium	[139]
	CPf1	Yield and Quality	Cuticular lipids	*Maize glossy2 gene*	Agrobacterium	[63]
	BEs			*ZmCENH3*	Agrobacterium	[137]

## Data Availability

Not applicable.

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
