# Peer review of "Evolution and Application of Genome Editing Techniques for Achieving Food and Nutritional Security"

_ijms, 2021, doi:10.3390/ijms22115585_

Round 1

Reviewer 1 Report

I suggest the following corrections:

Figure 1: The arrows start from 1985 and goes back to 1901 – please correct the timeline

Table 1: No mechanism of action for Cpf1/crRNA and base editing

2.1.2. CRISPR/Cpf1 system: Please mention about non-specific ssDNA cleavage activity of Cpf1.

3.1.1. CRISP – CRISPR

Line 258: Don’t italicize “the”

Figure 2: Figure 2 and its legend is not completely displayed. There are errors in depicting Cas9 and Cpf1 cleavage. Cas9 would cleave close to PAM and Cpf1 distal to the PAM site.

Line 445: Don’t italicize the whole line

Line 451: don’t italicize ‘encodes’

Line 550, 551, 552: Please checks where to italicize or not

Line 552, 553, 554: Sentence has been repeated with change of few words. Please concise.

Please write the references in uniform format

Author Response

Response to reviewers # 1:

Dear reviewer, we are grateful to you for your time and efforts for providing us with valuable comments and suggestions for the improvement of article under consideration. We have tried our level best to fix the revised article as per your comments and suggestion.

  1. Figure 1: The arrows start from 1985 and goes back to 1901 – please correct the timeline

Dear reviewer, we are grateful to you for your comments and suggestions.

Response: Thank you for your valuable comment for identification of problem and suggestion for improving the illustration. We have carefully looked into the timeline (Figure 1) and needful correction has been made based on events taken place in different years. The revised Figure 1 is incorporated in revised manuscript.   

  1. Table 1: No mechanism of action for Cpf1/crRNA and base editing.

Response: Thank you for your valuable comment to improve our manuscript. The genome editing technologies MegaN, ZFN, TALEN, CRISPR/Cas9, CRISPR/Cpf1 and base editing have same mechanism of action “Able to induce double strand breaks (DSB) with two possibilities of Non-homology end joining (NHEJ) and Homology directed repair (HDR), depends on designing tool”.  The submitted manuscript may be have formatting issue, showing Cpf1/crRNA and base editing mechanism of action was not described. The issue has been resolved in revised manuscript.

  1. 2.1.2. CRISPR/Cpf1 system: Please mention about non-specific ssDNA cleavage activity of Cpf1.

Response: Thanks for your worthy suggestions. The target-activated non-specific ssDNase activity, catalyzed by the same active site responsible for site-specific dsDNA cutting, is fundamental property of CRISPR/Cpf1 enzymes. The nuclease active site of Cpf1 cuts target single strand DNA (ssDNA) in cis and the non-target ssDNA in transposition. This nuclease can only embed one DNA strand at a time, so the target and non-target DNA strands are presumably cleaved sequentially. This sequential cleavage of DNA elucidates the mechanism of staggered-end DNA break induced by Cpf1 (Kleinstiver et al., 2016).

  1. 3.1.1. CRISP – CRISPR

Response: Thank you for your valuable comment. We have made correction in the revised manuscript.   

  1. Line 258: Don’t italicize “the”

Response: Thank you for your valuable comment. The needful correction has been made in the revised manuscript.

  1. Figure 2: Figure 2 and its legend is not completely displayed. There are errors in depicting Cas9 and Cpf1 cleavage. Cas9 would cleave close to PAM and Cpf1 distal to the PAM site.

Response: Dear reviewer, Thank you for your valuable comment for identification of problem and suggestion for improving the illustration. The needful correction in revised figure 2 have been made as per your suggestion. The figure caption along with abbreviations have been streamlined.

  1. Line 445: Don’t italicize the whole line

Response: Thank you for your valuable suggestion. The needful correction have been made in the revised manuscript and highlighted in red.

  1. Line 451: don’t italicize ‘encodes’

Response: Thank you for your valuable comment. The needful correction have been made in revised manuscript.

  1. Line 550, 551, 552: Please checks where to italicize or not.

Response: Thank you for your valuable comment. The needful correction have been made in revised manuscript.

  1. Line 552, 553, 554: Sentence has been repeated with change of few words. Please concise.

Response: Thank you for your valuable suggestion. The overlapping sentence has been removed for clarity of meaning.

  1. Please write the references in uniform format.

Response: Thank you for your valuable suggestion. The submission to IJMS is format free, we will made standard journal reference formatting once accepted.

Reviewer 2 Report

Thank you for providing me an opportunity to review this article, " Evolution and Application of Genome Editing Techniques for Achieving Food and Nutritional Security". 

1. The introduction is too long. Please keep in mind that the introduction should be precise, should have well-formulated research objectives, literature review very close to related studies, how the current review is different from the other closely associated studies, research gap, and significant findings. Simply this manuscript does not attempt to include all these aspects of the introduction. I would ask authors to follow recently published articles on how to write an introduction.

2. There have been tons of studies on, "Evolution and Application of Genome Editing Techniques for Achieving Food and Nutritional Security." How is this review different from the existing studies? What is their contribution?

3.  I would ask the authors to elaborate on how this study confirms and contradicts the existing literature with future recommendations.

4. Please introduce all acronyms, no matter how obvious, the first time they are used.

5. There are a few grammatical errors and spacing issues. Please make sure to take off. Please use the same font size.

6. So what is the follow-up? what needs to be investigated yet? What questions could be answered or should be addressed in future research?

Author Response

Response to reviewer # 2:

Dear reviewer, we are grateful to you for your comments and suggestions for the improvement of the article. We are thankful to your efforts and time for highlight key points to further strengthen the idea.

  1. The introduction is too long. Please keep in mind that the introduction should be precise, should have well-formulated research objectives, literature review very close to related studies, how the current review is different from the other closely associated studies, research gap, and significant findings. Simply this manuscript does not attempt to include all these aspects of the introduction. I would ask authors to follow recently published articles on how to write an introduction.

Response: Dear reviewer, thank you very much for your valuable comment on our manuscript. We have revised the introduction part as per your suggestions the objectives are clarified in the revised manuscript.

  1. There have been tons of studies on, "Evolution and Application of Genome Editing Techniques for Achieving Food and Nutritional Security." How is this review different from the existing studies? What is their contribution?

Response: Thank you for your valuable comment. Yes, we do agree with your worthy comment, many researchers have published their work on evolution, application and future trends of genome editing in plants and helped readership community to generate ideas based on their findings/literature. However, previous work/literature focused one or few points and readers have quest to get more detailed discussion evolution and application of GETs and other related techniques and their viable application to improve plants for traits of economic importance. The following review article has discussed all possible hotspots of research and proof of concepts with future direction to design and conduct experiments in plant sciences.  

  1. I would ask the authors to elaborate on how this study confirms and contradicts the existing literature with future recommendations.

Response: Thank you for your valuable comment on manuscript. The present review article is the continuation of previous work. The fast moving developments in plant sciences especially in biotechnology and its application in plant sciences needs continuous supply of information based on proof of concepts. The recent novel plant breeding techniques (NPBTs) with reference to genome editing has opened new avenues of research. We have tried to document the mechanism of GETs especially third generation, timeline during the development of GETs, detailed comparison of all three GETs, their application for yield and nutritional improvement, hybrid seed production especially “Multi-control sterility system” through multiplex genome editing, novel phenomena of induced apomixis to preserve hybrid vigor (benefit to reduce farmer cost of production as hybrid characteristic will remain preserved in multiple generations), transgene free breeding to skip regulatory regimes and integration of speed breeding to achieve genetic gain via genome editing. Based on these detailed discussion, we have concluded, GETs have potential to achieve food and nutritional security. We have added few lines in conclusion section of the revised manuscript.

  1. Please introduce all acronyms, no matter how obvious, the first time they are used.

Response: Thank you for your valuable comment/suggestion. We have gone through the whole manuscript (text, figures, tables) all abbreviations are described and full form is given at first place and subsequently abbreviation is used in the manuscript.

  1. There are a few grammatical errors and spacing issues. Please make sure to take off. Please use the same font size.

Response: Thank you for your valuable comment on manuscript. We have carefully revised the whole manuscript and needful correction are made throughout the manuscript.

  1. So what is the follow-up? what needs to be investigated yet? What questions could be answered or should be addressed in future research?

Response: Thank you for your valuable comment. The field is genome editing is evolving at very fast pace, the development of prime editing has been evaluated in few model plant species (rice, wheat and tomato) which need further investigation in model and non-model plant species. The induced apomixis through GETs (CRISPR/Cas9) has preserved hybrid vigour only 30 % in F2 population which need further investigation. Similarly, the multi-control sterility system is only tested in few model plant species which need to be tested and comparative studies are required among two-line and three-line hybrid seed development system. So far, the studies have concluded the success of multi-control system (one line hybrid) over three-line and two-line hybrid development system. In several countries, the genome editing crops are dealt same as GMOs, the advent of third generation GETs has enabled researchers to generate transgene free plants to skip strict regulatory regimes of several countries. However, there is still quest for a debate among all stakeholders to discuss all possible bio-safety concerns. The integration of omics, speed breeding and GETs has potential to achieve genetic gain which is essential to achieve food security. So far, these techniques are taken separately, we have proposed to utilized these techniques altogether to achieve zero hunger.   

Round 2

Reviewer 2 Report

Please work on spelling check, font size, and grammar. Thank you!